# Peritoneal Mesothelioma in a High Volume Peritoneal Surface Malignancies Unit

**DOI:** 10.3390/jcm12062288

**Published:** 2023-03-15

**Authors:** Fernando Pereira, Mónica Pereira, Israel Manzanedo, Ángel Serrano, Estibalitz Pérez-Viejo

**Affiliations:** 1Peritoneal Surface Malignancies Unit, Department of Surgery, Hospital Universitario de Fuenlabrada, 28942 Madrid, Spain; 2Faculty of Health Sciences, School of Medicine, Universidad Rey Juan Carlos, 28933 Madrid, Spain

**Keywords:** peritoneal mesothelioma, epithelioid subtype, cytoreductive surgery, HIPEC

## Abstract

Diffuse malignant peritoneal mesothelioma (PM) is a rare neoplasm, traditionally associated with a poor prognosis. There are other varieties of PM that are even less frequent and of uncertain malignancy. Cytoreductive surgery (CRS) with hyperthermic intraperitoneal chemotherapy (HIPEC) has achieved prolonged survival in selected patients. The aim of this study is to analyze the patients with PM assessed in our center. Clinicopathological characteristics, diagnostic procedures and survival results from patients with PM appraised at our unit, according to the applied treatment, were analyzed. Seventeen patients were assessed between 2007 and 2019. Three cases had multicystic PM that were treated with complete CRS + HIPEC; all patients are alive and free of disease after a long follow-up. Three other cases had biphasic PM; a curative treatment could be performed in none of them, and their survival was minimal (<6 moths). Lastly, 11 cases with epithelioid PM were treated. Two cases were considered unresectable at laparoscopy (PCI 39); one of them had a long survival (67 months) with three iterative laparoscopic palliatives HIPECs for refractory ascites. The other nine cases were treated with curative CRS + HIPEC, with a median PCI of 14 (range 4–25), and a median overall survival (OS) of 58 months, with a 5-year OS of 47.4%. In conclusion, CRS + HIPEC, when possible, appears to be the optimal treatment for patients with PM. Knowledge of this therapeutic option is crucial, both to offer it to patients and to avoid delays in their referral to appropriate centers for treatment.

## 1. Introduction

Diffuse malignant peritoneal mesothelioma (DMPM) is an aggressive neoplasm arising from mesothelial cells of the peritoneal serosa, and may affect the peritoneal surface more or less extensively. The incidence of DMPM is very low (0.2–3 cases/million inhabitants/year) [1], much lower than that of pleural mesothelioma [2], and less related to asbestos exposure than the pleural variety [1]. The most common histological subtype is epithelioid (75% of cases), followed by sarcomatoid (13%) and biphasic (6%) [3]. Traditional treatment of DMPM is systemic chemotherapy (CT) with palliative surgery on demand, resulting in a fatal prognosis. The median overall survival (OS) with contemporary systemic CT (pemetrexed + platinum) [4] is about 12 months, with response rates around 30%. The implementation of cytoreductive surgery (CRS) with hypertermic intraperitoneal chemotherapy (HIPEC), especially in the last two decades, has been a therapeutic milestone, reaching survival rates of 53 months (range 34–92 months) with a 5-year OS of 47% [5]. Nowadays, it is considered the treatment of choice for all those patients in whom complete cytoreduction seems possible. Careful patient selection and center experience are essential to optimize both postoperative and long-term survival outcomes [6,7]. HIPEC has also been successfully used as a palliative treatment for refractory ascites in patients with unresectable disease [8,9].

There are two other varieties of diffuse peritoneal mesothelioma (PM) that are considered borderline (well-differentiated papillary and multicystic), as they can relapse after surgery and exceptionally progress to DMPM [10,11]. In these varieties, CRS + HIPEC is considered a better alternative than isolated CRS [7].

Due to the low incidence of the disease, there are very few centers with large series of PM treated with CRS + HIPEC [12,13]. In Spain, we have only found a small series (7 patients) published in 2007 [14]. However, two essential multi-center studies [5,15], a meta-analysis [16] and multiple reviews have been published, among which it should be highlighted the recent PSOGI/EURACAN guideline [7].

The aim of this study is to analyze the outcome of patients with PM assessed at our high volume Peritoneal Surface Malignancies Unit (in which we perform around 100 annual CRS + HIPEC procedures for different indications, including PM).

## 2. Materials and Methods

Patients with PM referred to our Peritoneal Surface Malignancy Unit for diagnostic and/or therapeutic evaluation were analyzed. Treatment options were evaluated in the Multidisciplinary Tumor Board (MTB), taking into account the performance status, the histological variety and the extent of the disease in imaging test or by laparoscopy, including the indication of preoperative CT (pemetrexed-platinum) in potentially resectable cases with malignant varieties.

Patients for whom surgery was considered to be beneficial were operated with initial curative intention, although the final decision on cytoreduction was made during the intervention, after a precise assessment of the extent of the disease. CRS was performed with a variety of selective peritonectomies and visceral resections, and it was at least attempted that residual disease remain millimetric in those cases in which complete cytoreduction was not possible. In patients in whom significant cytoreduction was achieved (CCS 0-1), HIPEC with Cisplatin 100 mg/m^2^ + Doxorubicin 15 mg/m^2^ was carried out for 90 min. In two of these patients, during the early years of our program, EPIC (early postoperative intraperitoneal chemotherapy) was also used during the first 3–5th postoperative days (with paclitaxel 20 mg/m^2^ in 1000 cc of peritofundin). Finally, in cases where the disease was considered unresectable, palliative HIPEC was carried out (laparoscopically when possible) in those with malignant ascites.

Major postoperative complications were recorded (Dindo–Clavien classification [17]). Patients were reevaluated again by the MTB after discharge, and a final decision was made on the need of postoperative CT and the follow-up protocol. In case of recurrence, patients were treated according to a new decision of the MTB, even with surgery (including the possibility of additional CRS + HIPEC) when the appropriate criteria were met.

Informed consent was obtained from all individuals included in this study. The research has been approved by the authors’ institutional review board.

Statistical study: Qualitative variables are described with their distribution frequencies. Quantitative variables are described with their medians and ranges. The Kaplan–Meier method was used for survival analysis. All statistical analyses were performed using SPSS 25.0.

## 3. Results

From 2007 to 2019, 17 patients with PM were assessed in our Peritoneal Surface Malignancies Unit. Data were analyzed in February 2020. The clinical characteristics are summarized in Table 1. Only five patients were initially evaluated in our own center, while the rest came from other centers in Madrid (*n* = 7) or other regions of Spain (*n* = 5).

Three of the cases were women with multicystic PM, all initially asymptomatic and discovered incidentally at gynecological examinations. In one of them, two previous incomplete cytoreductions had been performed in another center. All three were treated with complete CRS (CCS 0) + HIPEC, with a median PCI (Peritoneal Cancer Index) of 18 (range 8–21). None had serious complications (only one had a minor complication consisting of low-grade fever with no clear focus), and the median hospital stay was 7 days (range 7–13). After a median follow-up of 59 months (range 33–127), all remain alive and free of disease.

Three other cases correspond to biphasic DMPM. One 83-year-old patient was diagnosed at an urgent and palliative surgery for bowel obstruction, with post-operative death after 5 days. The other two patients presented with progressive abdominal distension and constitutional syndrome. One of them was admitted to our center for study and a laparoscopy was performed, declaring the tumor unresectable (PCI 30). The last patient was diagnosed (by laparoscopy) and received CT (pemetrexed + cisplatin) in other hospital; in our center, a second restaging laparoscopy was performed and an attempt of CRS + HIPEC was made, but it was considered finally unresectable (PCI 39), performing a palliative HIPEC for malignant ascites. Both patients received palliative CT but had minimal survival (6 months). 

Finally, 11 cases with epithelioid DMPM have been treated. All debuted with variable patterns of distension and/or abdominal pain and/or constitutional syndrome, except for one asymptomatic case diagnosed after the removal of an umbilical nodule (suspected umbilical hernia).

In two finally unresectable cases (both women), the diagnosis of extensive disease was made at an initial laparoscopy, followed by neoadjuvant CT (NACT), both being definitely unresectable (PCI 39) after a second restaging laparoscopy. In one of them, palliative HIPEC was applied (in the 2nd laparoscopy) for the treatment of ascites. This patient (in whom laparoscopic HIPEC was subsequently repeated twice for refractory ascites) had a long survival (67 months). The other patient was lost after 4 months of follow-up (she came from outside Madrid). 

In the other nine cases of epithelioid DMPM, CRS + HIPEC was performed with curative intent (CCS 0 in 7, CCS 1 in 2). In 7/9, a previous staging laparoscopy had been performed and five of them received NACT for extensive disease. In addition, the remaining two patients who did not undergo staging laparoscopy had also received NACT previously in other hospitals (a total of seven patients with NACT) and they were referred to our center after confirming response (in one) or stable disease (in the other), being then operated without previous laparoscopy, both with low PCI (6 and 4) at surgery. Median surgical PCI in the nine patients was 14 (range 4–25) with the following distribution: PCI < 10 in 3 cases, PCI 10–20 in three cases, and PCI > 20 in three cases. The peritoneal/visceral resection procedures are detailed in Table 2.

The median duration of the nine CRS + HIPEC procedures was 360 min (range 300–510). Two patients also received EPIC. One patient died on the 11th postoperative day after extensive (PCI 25) and complete cytoreduction (CCS 0) with politransfusion, due to multiorgan failure secondary to systemic inflammatory response syndrome and sepsis, without surgical complications. Only one other patient had serious complications (grade III Dindo-Clavien) with organ-space SSI and reoperation for intestinal leak and evisceration. The rate of severe complications was 22.2% (2/9), including the exitus. Another four patients had minor complications (acute urinary retention, ileus, urinary tract infection, seroma), with a total of complications of any grade of 66.6% (6/9). The median hospital stay was 11 days (range 6–30). 

Adjuvant CT was administered in four of the seven patients who had received NACT, all after extensive cytoreductions (PCI > 13). Two patients who received NACT but had low surgical PCI (6 and 4) did not receive adjuvant CT. Only one patient with high PCI (16), who had received NACT, did not receive postoperative CT due to poor postoperative performance status.

Of the eight patients in whom CRS was possible with curative intent and who survived the intervention (excluding the postoperative exitus), five have relapsed (four in the peritoneum and one axillary lymph node recurrence). Surgical rescue was attempted in all five, but 3/4 peritoneal recurrences were considered unresectable at re-laparotomy (in one patient on two occasions); the planned HIPEC was also ruled out in two of them (twice in one of these patients) and two palliative HIPECs were performed in the third (with ascites and imprecise PCI for encapsulating peritoneal sclerosis), who is still alive despite persistent disease 110 months after the first surgery (very prolonged OS). In the other two relapses (one peritoneal and the other axillary lymphatic), complete resection was achieved without subsequent recurrence (with CRS + HIPEC in the first and bilateral axillary lymphadenectomy in the latter, although this one has died in the follow-up due to another cause) (Figure 1). 

Perioperative data of all CRS + HIPEC performed with curative intention, three in multicystic PM and ten in epithelioid DMPM (9 at initial presentation and 1 at relapse) are summarized in Table 3.

With a median follow-up of 49 months, the median OS in the nine patients with epithelioid DMPM in whom CRS with curative intent was possible (including the postoperative exitus) is 58 months, with a 5-year OS of 47.4%. The median disease-free survival (DFS) is 17 months, with a 4-year DFS of 38% (Figure 2).

In total, we programmed 22 HIPECs in 14 patients, four of which had initial planned-palliative intention (one first, one second and two third HIPECs). Finally, of the 18 attempts of CRS + HIPEC with curative intent (13 in first attempt, 4 in second attempt and 1 in third attempt) only 13 were curative (12/13 in first attempt, 1/4 in second attempt and 0/1 in 3rd attempt). Therefore, CRS was aborted in 5/18 (27.7%) attempts of CRS + HIPEC with curative intent, and the number of palliative HIPECs increased from 4 to 6 (as an unplanned-palliative HIPEC was performed in 2 of these 5 aborted-CRS cases). In summary, we finally performed a total of 19 HIPECs (6 palliative and 13 with curative intent). The six palliative HIPECs have been performed in three patients, and three of them have been carried out by laparoscopy (all in the same patient).

A total of 18 of the 22 scheduled HIPECs were planned in 10 epithelioid DMPM (Figure 1), 4 of which had initial planned-palliative intention (one first, one second and two third HIPECs). Finally, of the 14 attempts of CRS + HIPEC with curative intent (9 in first attempt, 4 in second attempt and 1 in third attempt), only 10 were curative (9/9 in first attempt, 1/4 in second attempt and 0/1 in 3rd attempt). Therefore, CRS was aborted in 4/14 (28.5%) attempts of CRS + HIPEC with curative intent in epithelioid DMPM, and the number of palliative HIPECs in epithelioid DMPM increased from 4 to 5 (as it was performed a 2nd unplanned-palliative HIPEC in 1 of the 3 aborted-CRS cases with finally unresectable peritoneal relapse).

## 4. Discussion

Due to the low incidence of PM, it is of utmost importance to concentrate patients in centers with expertise in the treatment of peritoneal diseases, in which the learning curve (both in the selection of patients and in the highly complex surgical procedures) no longer has a negative impact on the outcomes. Of the seventeen patients with PM assessed at our center, only five were initially evaluated in our own hospital, while the rest were referred from other centers.

Precise patient selection for CRS + HIPEC is crucial in DMPM to avoid unnecessary laparotomies and save surgical resources. Thus, it is highly recommended to use laparoscopic staging whenever possible [18], even though there still is a risk of underestimating the real extension of the disease [19]. In our patients, staging laparoscopy was performed on 2/3 patients with biphasic PM (not on the one diagnosed in urgent surgery for bowel obstruction), and on 9/11 patients with epithelioid PM. In some patients (one biphasic and two epithelioid), even two laparoscopies were carried out, one diagnostic of the unresectable mesothelioma and another one after neoadjuvant chemotherapy (NACT) to reevaluate a chance of cytoreduction. 

Initial laparoscopies allowed the exclusion of three patients for CRS (in one of them palliative laparoscopic HIPEC was performed), and of the first 13 surgeries scheduled with a curative intent, only one was suspended (a biphasic PM in which there seemed to be a possibility of cytoreduction at laparoscopy after NACT, but a palliative HIPEC was finally performed). However, of the five curative-intent surgeries scheduled for relapses (4 in second and 1 in third attempts), four CRS had to be discarded due to intraoperative irresectability (one with palliative HIPEC). In these cases, the role of laparoscopic assessment is very limited (or impossible) due to previous extensive open surgery, and there is no alternative but to estimate the possibility of cytoreduction based on imaging tests. Overall, 5 of the 18 attempts of CRS + HIPEC with a curative intent were aborted (38.4%), a result that is consistent with the rate of incomplete CRS reported in the literature (33%, range 7–57%) [16].

CRS + HIPEC procedures in specialized centers are associated with a mortality rate of 0.9–5.8% and serious perioperative morbidity of 12–52% [20]. In our PM series, there was one post-operative exitus among the 13 CRS + HIPEC procedures. This mortality is high (1/13 = 7.6%), but no conclusions can be drawn from such a small series. In our overall series of CRS + HIPEC for any indication (including colon, gastric, ovarian cancer, peritoneal pseudomyxoma, PM and non-conventional indications), currently exceeding 900 cases, postoperative mortality is 3%, similar to that of most expert centers. However, serious morbidity (Dindo-Clavien III–V, including the postoperative exitus) of this series is low, only present in 2 of the 13 procedures (15.3%) (Table 3). 

Different cytotoxic drugs have been used for HIPEC in PM, mainly cisplatin and mitomycin C, administered alone or in combination with doxorubicin or other drugs. It seems that the best result is obtained with combined schemes [21], based on platinum at least when CRS is complete [15]. 

The use of NACT in DMPM is under debate, and there are even authors who consider it harmful when a complete CRS can be achieved [22,23]. It is usually administered when there are doubts about resectability. In our series, 7/9 patients with epithelioid DMPM treated with complete CRS + HIPEC had received NACT with pemetrexed-cisplatin. The recent international recommendations of PSOGI/EURACAN specify three scenarios: (1) resectable patients, (2) clearly unresectable patients, and (3) borderline resectable patients [7]. Primary CRS + HIPEC is considered the treatment of choice when the disease is resectable. In patients with unresectable or borderline disease, there is the option of neoadjuvant CT (even bidirectional with intraperitoneal pemetrexed + intravenous cisplatin), having reported surgical rescues in up to 50% of cases [24]. In this regard (conversion to resectability), great expectations have recently been raised with the use of PIPAC (pressurized intraperitoneal aerosol chemotherapy) in different indications [25], and a specific trial in DMPM has even been designed [26].

The use of adjuvant CT after CRS + HIPEC is considered beneficial, especially in the presence of any adverse prognostic factor (CCS ≥ 1, sarcomatoid or biphasic subtypes, lymph node metastases, high PCI or Ki-67 > 9%) [7]. Nevertheless, adjuvant CT can be avoided in patients with favorable prognosis. In our series, five of the nine patients with epithelioid PM treated with complete CRS + HIPEC did not receive postoperative CT. However, this includes two cases that would have been candidates but did not receive it for other reasons: the postoperative exitus and another patient with poor postoperative performance status.

Only a few multi-institutional registries have managed to gather a large number of patients with PM treated with CRS + HIPEC. The best known are those published by Yan et al. in 2009 [5] (405 patients with a median OS of 53 months and 5-year OS of 47%) and Alexander et al. in 2013 (211 patients from three US institutions with a median OS of 38.4 months and 5-year OS of 41%). A meta-analysis by Helm et al. in 2015 includes 1047 patients from 20 studies with a 5-year OS of 42% [16]. Our results in the nine patients with epithelioid PM in whom CRS + HIPEC has been possible with curative intent (median OS 58 months, 5-year OS 47.4%) are therefore in the high range of the reference series. Our DFS (median 17 months, 4-year 38%), although also remarkable, underestimates the real benefit since it is calculated up to the date of the first relapse; however, a complete cytoreduction was achieved in two of the recurrences (with no subsequent relapse). Therefore, the patients free of disease at the time of data analysis are not 3 out of 9, but 5 out of 9; however, this number is not reflected in the DFS concept.

The sarcomatoid subtype has such a poor prognosis that it is actually considered a contraindication for CRS + HIPEC [7]. The prognosis of biphasic PM is worse than the epithelioid subtype; however, it is possible to increase the survival rate with complete CRS + HIPEC in properly selected patients [27]. Our study clearly demonstrates the worst prognosis of biphasic tumors, with unresectable disease and survival of less than 6 months in the three cases. 

Palliative HIPEC can be used in DMPM for the treatment of malignant ascites [8,9]. In our study, six palliative HIPECs were performed (three of them laparoscopic) in three patients, with prolonged survival in two of them (both epithelioid). PIPAC, combined with systemic CT, has been used in peritoneal carcinomatosis of various origins mainly with palliative intention, leading to clinical response rates of 67–75% in DMPM [28]. However, it has never been compared with laparoscopic palliative HIPEC, which is ostensibly cheaper and surprisingly forgotten in favor of PIPAC.

Multicystic PM has a much better prognosis, is more common in women (83%), and usually has an incidental diagnosis, as reflected in our series. Traditional treatment is surgical resection, although long-term follow-up is necessary due to the high probability of recurrence (50%) and the exceptional possibility of malignant transformation [11]. CRS + HIPEC in experienced centers is considered the treatment of choice nowadays [7], and the three patients in our series (treated with CRS + HIPEC without postoperative mortality or serious complications) all remain alive and free of disease after a very long follow-up (59 months).

## 5. Conclusions

Our data confirm that treatment of PM with CRS + HIPEC, in correctly selected patients, seems to be the optimal treatment. It is important to know this therapeutic option, both to offer it to patients, and to avoid delays in referral to appropriate centers for treatment. CRS + HIPEC is considered a better alternative than isolated CRS also in borderline varieties of diffuse PM (well differentiated papillary and multicystic). Candidate selection should include laparoscopic staging whenever possible in DMPM. Perioperative systemic chemotherapy is indicated in certain cases, and therapeutic decisions should be made in a Multidisciplinary Tumor Board in expert centers in the treatment of peritoneal surface malignancies. Patients with unresectable disease may benefit from the use of palliative (preferably laparoscopic) HIPEC for the treatment of malignant ascites.

## Figures and Tables

**Figure 1 jcm-12-02288-f001:**
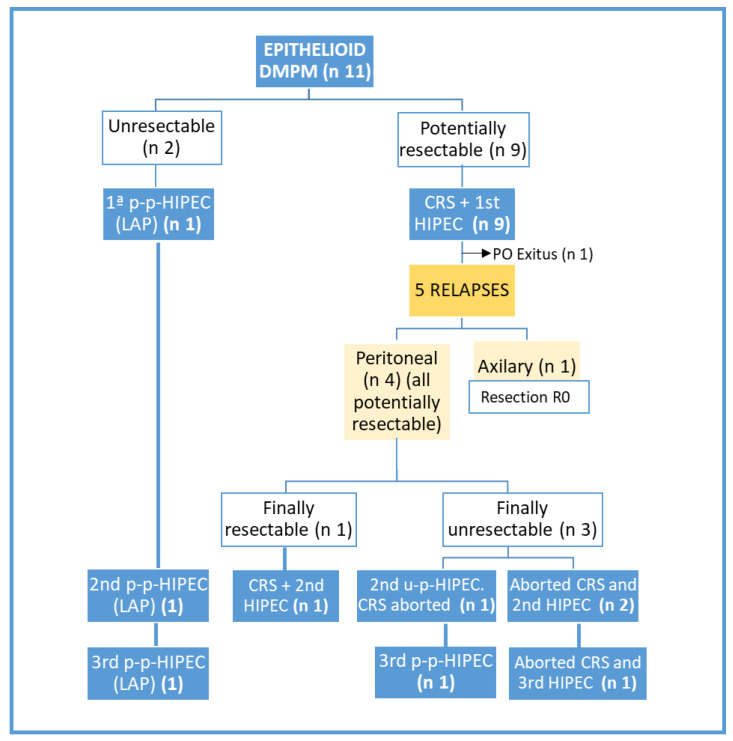
Flowchart of programmed, performed, palliative/curative and aborted HIPECs in epithelioid DMPM. “DMPM”: diffuse malignant peritoneal mesothelioma; “CRS”: cytoreductive surgery; “p-p−HIPEC”: planned-palliative HIPEC; “u-p−HIPEC”: unplanned-palliative HIPEC; “LAP”: laparoscopic; “PO Exitus”: postoperative exitus.

**Figure 2 jcm-12-02288-f002:**
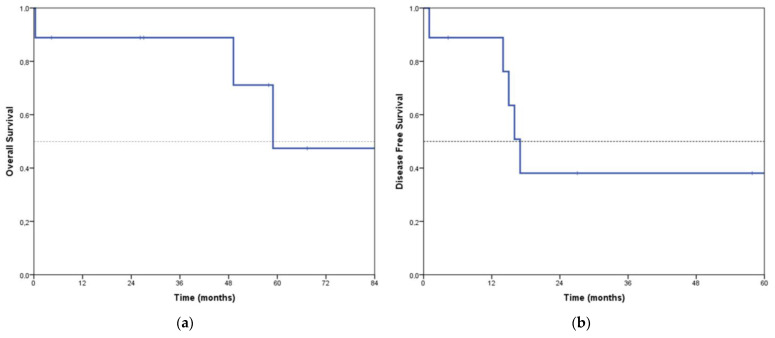
Overall survival (**a**) and disease−free survival (**b**) in the nine epithelioid peritoneal mesotheliomas with complete CRS + HIPEC.

**Table 1 jcm-12-02288-t001:** Descriptive statistics.

	Multicystic	Biphasic	Epithelioid
N° of patients	3	3	11
Origin: our center/other centers	1/2	2/1	2/9
Age (median)	51	64	57
Sex (female/male)	3/0	1/2	5/6
Asbestos exposure	0	0	1
Previous abdominal surgeries	1	0	1
Unresectable	0	3	2

**Table 2 jcm-12-02288-t002:** Resections performed in epithelioid PM cases with complete cytoreduction (*n* = 9).

Procedures	Number
Major omentectomy	9
Appendectomy	7
Right diaphragm peritonectomy	6
Left diaphragm peritonectomy	4
Morrison peritonectomy	4
Hepatoduodenal ligament	4
Lateral parietal peritonectomy	4
Pelvic peritonectomy	4
Cholecystectomy	4
Splenectomy	3
Anterior parietal peritonectomy	3
Right hemicolectomy	3
Total hysterectomy	1
Bilateral salpingo−oophorectomy	1
Anterior resection of rectum	1
Superior recess of the omental bursa	1
Hepatic capsulectomy (partial)	1
Small bowel resection	1
Electrofulgurations *	5

* mesenteric or left diaphragm.

**Table 3 jcm-12-02288-t003:** Perioperative data of all curative CRS + HIPEC for peritoneal mesothelioma.

	Multicystic (n = 3)	Epithelioid (n = 10)	Total (n = 13)
Staging laparoscopy	0	7	*
Neoadj treatment	NA	7	*
surgical PCI, median (range)	18 (8–21)	14 (4–25)	15 (4–25)
Duration (min), median (range)	350 (240–350)	360 (300–510)	360 (240–510)
Transfusion	0	1	1
Complic grade ≥ III	0	2	2 (15.3%)
Complic any grade	1	6	7 (53.8%)
Reoperation	0	1	1
Mortality	0	1	1 (7.6%)
Length of stay (days), median (range)	7 (7–13)	11 (6–30)	10 (6–30)
Adjuvant treatment	NA	4	*

“NA”: not applicable. * Not recorded since, in multicystic PM, there is no indication for systemic chemotherapy.

## Data Availability

All data supporting reported results are provided as part of the manuscript, and no new datasets were created.

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
