# Peer review of "Peritoneal Mesothelioma in a High Volume Peritoneal Surface Malignancies Unit"

_jcm, 2023, doi:10.3390/jcm12062288_

Round 1
Reviewer 1 Report
I would like to thank the authors for their study presentation.
English language and overall: good
The topic is not a new one, representing the experience of one center with very few numbers of cases.
Title:
· the authors wrote “HIGH VOLUME PERITONEAL SURFACE MALIGNANCY UNIT”>> what do you mean by “peritoneal surface malignancy unit”>> do you mean “malignancies” or just mesothelioma, kindly elaborate and modify.
Abstract:
· What are the risk factors in those patients (eg asbestos?) and mean survival rate of all the cohort?
Introduction:
· Diffuse Malignant Peritoneal Mesothelioma>> please mention the incidence and prevalence, also and survival rate with and without treatment in the literature as compared to other types of less aggressive peritoneal malignancies.
Methods:
· Could you add another flow chart of how the patients were managed according to the years or modify the current one, as the guidelines changed you changed some of the chemotherapy as apparent in the text, but a flow diagram will be easier to follow, and also please add references to the guidelines.
· Was there any neoadjuvant medications or genetic marker study done for the studies cohort?
Results:
· You mentioned the side effects in the methodology “Major postoperative complications were recorded (Dindo-Clavien classification [17]). “>>>but no table presentation is found in the manuscript, kindly add.
Discussion:
The authors stated “However, seri-224 ous morbidity (Dindo-Clavien III-V, including the postoperative exitus) of this series is 225 low, only present in 2 of the 13 procedures (15.3%).”>> please mention the type of adverse events encountered and how you dealt with them, and if they had any effect on mortality.
Conclusion:
Kindly add a statement on the effect of different types of tumors and chemotherapeutic agents on survival.
References: good, although I suggest adding the metanalysis and guidelines references on the topic.
Reviewer 2 Report
Thank you for the opportunity to review the manuscript entitled (PERITONEAL MESOTHELIOMA IN A HIGH VOLUME 2 PERITONEAL SURFACE MALIGNANCY UNIT). This manuscript is interesting and focuses scientific attention on an important topic. Although the presented data are sound, there are a few points that have to be addressed
My comments:
The title needs revision so that it indicated the value of the work
The number of patients is merely sufficient to the conclusions (i.e limited number was used)
Reviewer 3 Report
Pérez-Viejo et al have presented a well conducted presentation of cases of peritoneal mesothelioma at their centre. This is of much value in the field of epidemiology as well as for oncologists in Spain and neighbouring regions. The authors refer to the PSOGI/EURACAN clinical practice guideline for diagnosis, treatment and follow-up and their results are in support of the guidelines.
As newer treatments evolve, it would be interesting to report in future the rates of PD-L1 positivity and results from immunotherapy treatment regimes in peritoneal mesothelioma. Comparing survival rates to with these recent treatments will make a significant contribution to policy and treatment expectations.
Round 2
Reviewer 2 Report
Thanks for authors to revise this manuscript carefully. I think this revised manuscript can be published.